# Acidocin A and Acidocin 8912 Belong to a Distinct Subfamily of Class II Bacteriocins with a Broad Spectrum of Antimicrobial Activity

**DOI:** 10.3390/ijms251810059

**Published:** 2024-09-19

**Authors:** Daria V. Antoshina, Sergey V. Balandin, Ekaterina I. Finkina, Ivan V. Bogdanov, Sofia I. Eremchuk, Daria V. Kononova, Alena A. Kovrizhnykh, Tatiana V. Ovchinnikova

**Affiliations:** 1M.M. Shemyakin and Yu.A. Ovchinnikov Institute of Bioorganic Chemistry, Russian Academy of Sciences, 117997 Moscow, Russia; riruka11@mail.ru (D.V.A.); finkina@mail.ru (E.I.F.); contraton@mail.ru (I.V.B.); sofiia.eremchuk@gmail.com (S.I.E.); kononova.dv@phystech.edu (D.V.K.); potemkina.aa@phystech.edu (A.A.K.); ovch@ibch.ru (T.V.O.); 2Moscow Center for Advanced Studies, 123592 Moscow, Russia; 3Department of Biotechnology, I.M. Sechenov First Moscow State Medical University, 119991 Moscow, Russia

**Keywords:** antimicrobial peptides, bacteriocins, *Candida albicans*, lactic acid bacteria, *Lactobacillus acidophilus*, membranolytic peptides, pediocin box, transepithelial transport

## Abstract

Within class II bacteriocins, we assume the presence of a separate subfamily of antimicrobial peptides possessing a broad spectrum of antimicrobial activity. Although these peptides are structurally related to the subclass IIa (pediocin-like) bacteriocins, they have significant differences in biological activities and, probably, a mechanism of their antimicrobial action. A representative of this subfamily is acidocin A from *Lactobacillus acidophilus* TK9201. We discovered the similarity between acidocin A and acidocin 8912 from *Lactobacillus acidophilus* TK8912 when analyzing plasmids from lactic acid bacteria and suggested the presence of a single evolutionary predecessor of these peptides. We obtained the *C*-terminally extended homolog of acidocin 8912, named acidocin 8912A, a possible intermediate form in the evolution of the former. The study of secondary structures and biological activities of these peptides showed their structural similarity to acidocin A; however, the antimicrobial activities of acidocin 8912 and acidocin 8912A were lower than that of acidocin A. In addition, these peptides demonstrated stronger cytotoxic and membranotropic effects. Building upon what we previously discovered about the immunomodulatory properties of acidocin A, we studied its proteolytic stability under conditions simulating those in the digestive tract and also assessed its ability to permeate intestinal epithelium using the Caco-2 cells monolayer model. In addition, we found a pronounced effect of acidocin A against fungi of the genus Candida, which might also expand the therapeutic potential of this bacterial antimicrobial peptide.

## 1. Introduction

Bacteriocins, ribosomally synthesized antimicrobial peptides (AMPs) of bacterial origin, are currently the subject of considerable interest from both basic biology and biotechnology [1,2,3]. Over the years of study, various approaches to the classification of bacteriocins have been proposed based on their natural origin, biosynthesis, physicochemical properties, structure, antimicrobial activity and mechanisms of action [4,5]. At present, the most common is the structural classification, according to which bacteriocins are subdivided into four classes. Classes I and II are small (less than 10 kDa) thermostable peptides that undergo significant posttranslational modifications (class I) or are devoid of them (class II). Class III includes larger thermolabile proteins with molecular masses greater than 10 kDa, which are divided into two subclasses: bacteriolysins, enzymes capable of degrading the peptidoglycan of the bacterial cell wall, and non-lytic bacteriocins. Class IV bacteriocins include other large proteins with molecular masses from 20 to 100 kDa, mainly produced by Gram-negative bacteria: colicins and colicin-like bacteriocins, as well as tailocins, which resemble the tail of bacteriophages [1,3,4,5,6]. Most of the known class I and II bacteriocins have been isolated from Gram-positive bacteria and are active against closely related species [3]. The main protective factor providing resistance of Gram-negative bacteria to class I and II bacteriocins is the outer membrane. The few bacteriocins that can overcome this barrier include enterocin HDX-2 [7], plantaricins ZJ5 [8] and JLA-9 [9], pediocin IE-3 [10], bacteriocins LSX01 [11] and YD1 [12], and lactocin C-M2 [13].

Among class II unmodified bacterial AMPs, pediocin-like bacteriocins (PLBs, the subclass IIa bacteriocins) are particularly numerous. These peptides are characterized by the presence of a so-called pediocin box—a conserved *N*-terminal “YGNG(V/L)” motif, usually followed by a small 6-residue ring CXXXXC formed by a disulfide bond. They typically have a narrow spectrum of antimicrobial activity, with each PLB exhibiting its effect in submicromolar concentrations against a subset of Gram-positive bacterial species closely related to the producer strain. PLBs have potential for use in the food industry due to their specific activity against food spoilage strains and pathogens [14,15,16]. Some PLBs are considered new effective antimicrobial agents for application in medicine as they are active against *Listeria* and *Enterococci* [3,17,18].

We have previously shown [19] that acidocin A, the PLB with the non-canonical structure produced by the food strain *Lactobacillus acidophilus* TK9201 [20], had a wide spectrum of antibacterial activities and was capable of inhibiting both Gram-positive and Gram-negative bacteria, including pathogenic ones, which significantly distinguished it from the classical representatives of PLBs. The pediocin box in acidocin A did not appear to carry an important functional load since introduced mutations and even complete deletion of this part of the sequence only slightly reduced antimicrobial activities against the strains tested [19]. At the same time, we found out that acidocin A and a classical PLB avicin A demonstrated strong immunomodulatory effects on primary human monocytes.

When searching for putative bacteriocin genes in nucleotide sequence databases, we discovered a sequence encoding the *C*-terminal part of acidocin A in the 3′-untranslated region, immediately after the stop codon, of the gene of acidocin 8912, the bacteriocin from the *Lactobacillus acidophilus* TK8912 strain [21,22]. Some similarity with acidocin 8912 was also found in the *N*-terminal part of mature acidocin A (Figure 1). It is of note that some characteristic features of the pediocin box sequences of acidocin A and bacteriocin OR-7 from *Ligilactobacillus salivarius* NRRL B-30514, including the threonine insertion, are also present in the structure of acidocin 8912.

In this work, we performed further functional studies of acidocin A with the use of avicin A as a reference peptide, including analysis of their antifungal activities, proteolytic stabilities and transepithelial permeabilities, to better understand the biological role and evaluate the therapeutic potential of acidocin A. We also compared its activity spectrum with that of acidocin 8912, which has remained understudied so far, and with the recombinant peptide comprising the sequences of acidocin 8912 and the *C*-terminal acidocin A-like extension (“acidocin 8912A”, Figure 1)—a possible evolutionary predecessor of acidocin 8912. Acidocin A confirmed its reputation as an AMP with a broad spectrum of activity. The effects of acidocin 8912 were shown to be weaker, but its activity spectrum indicated that both peptides belong to the same functional group.

## 2. Results

### 2.1. Acidocin 8912 as a Homolog of Acidocin A

BLAST search for acidocin A through nucleotide sequences of bacterial origin in NCBI nr/nt database resulted in six entries producing significant alignments: D43626.1, CP029547.1, FM246455.2, CP127402.1, AB081463.1, and HQ259052.1. The first one is the gene of acidocin 8912—a class II bacteriocin from *Lactobacillus acidophilus* TK8912—which has not yet been assigned to any of the known AMP families [22]. Other entries refer to plasmids from *Lactobacilli* and related genera containing the same gene for acidocin 8912. In each case, a sequence encoding the *C*-terminal part of acidocin A is present downstream of the stop codon that terminates the translation of acidocin 8912 (Figure 1). Acidocin A and acidocin 8912 share the same secretion signal with a double glycine cleavage site for the type I secretion system (ABC transporter). It is worth noting that there is a slight similarity between acidocin 8912 and the *N*-terminal part of acidocin A (31% of identical residues), which escapes attention if we compare mature peptides without considering the genetic context. Acidocin 8912 lacks the first cysteine residue of the acidocin A sequence. The structures of both acidocins and OR-7 contain a conserved tryptophan residue. Remarkably, the alignment of the first 17 residues does not contain gaps.

### 2.2. Expression and Purification of the Recombinant Peptides

The peptides used in this work do not undergo the post-translational modifications, except for disulfide bond formation, so they were obtained using *E. coli* BL21(DE3) expression system based on T7 promoter and His-tagged thioredoxin as the carrier protein. The peptides were purified using reversed-phase HPLC (Appendix A), and all the measured *m*/*z* values matched the corresponding calculated molecular masses (Appendix A). The purity of the peptides, as assessed by Tris-tricin SDS-PAGE, was ≥95% (≥90% for acidocin 8912A, which had an elution time close to His8-thioredoxin). The final yields were in the range of 1–6 mg per 1 L of bacterial culture. The low yield of acidocin 8912A (about 1 mg/L) is probably due to its susceptibility to proteolysis because of its extended structure, which is not stabilized by a disulfide bond (as in the case of acidocin A) and is partly of artificial origin.

As part of structure–activity relationship studies, we obtained a set of shortened analogs of acidocin A (Figure 2) as well as avicin A-acidocin A hybrids (Appendix A).

### 2.3. Secondary Structure of Acidodin 8912 and Acidocin 8912A

The secondary structure of acidocin 8912 and acidocin 8912A was examined by CD spectroscopy (Figure 3, Table 1) under the same conditions as the structure of acidocin A and avicin A in our previous study [19]. If we compare the data for all four peptides, we can see that acidocin 8912 in aqueous solution is distinguished from the others by a particularly high content of β-folded structure, the proportion of which predominates over disordered regions. In the presence of detergent micelles, the proportion of α-helices increases dramatically for all peptides, but it is much more pronounced for acidocins (40–50%) than for avicin A (about 20%). This correlates with the higher membranolytic activity of acidocins compared to classical PLBs. Formation of α-helices in avicin A occurs primarily through a decrease in disordered regions, whereas in acidocins, it occurs primarily through a decrease in the proportion of β-folded regions. Acidocin 8912 is the only one of the four peptides that differentially changes its conformation in the presence of anionic (SDS) and non-ionogenic (DPC) detergents.

### 2.4. Antibacterial Activity

This work provides new data on the antibacterial activity of acidocin A and acidocin 8912, as well as information on the activity of acidocin 8912A and the truncated analogs of acidocin A (Table 2 and Table 3).

In terms of clinical relevance, of particular interest is the high activity of acidocin A against vancomycin-resistant strains of *E. faecium*, as well as against the strain of *M. smegmatis*, a model microorganism related to the causative agent of tuberculosis.

The activity of acidocin 8912 was lower than that of acidocin A in all tests—the difference was manifested to different degrees in different strains. Nevertheless, the peptide demonstrated a wide spectrum of activity, including Gram-positive bacteria of the genera *Lactococcus*, *Enterococcus*, *Bacillus*, *Mycobacterium* and *Micrococcus*, as well as several strains of *E. coli* (Gram-negative bacteria). Acidocin 8912A has only been tested on a limited panel of strains, but this peptide also showed the ability to inhibit both Gram-positive and Gram-negative bacteria at approximately the same concentrations as acidocin 8912. The lack of clear advantages over other peptides, the large size of the molecule, and the difficulty in obtaining the preparation discouraged us from its more extensive testing. Avicin A, as previously [19], did not show activity in serial dilution assays, except for the test on *Listeria*. It should be noted that in agar diffusion assay, avicin A demonstrated effective inhibition of all three *E. faecium* vancomycin-resistant strains, which was also observed during the initial stage of incubation in a liquid medium. However, after 24 h incubation in the liquid medium, overgrowth of the well by the test culture reproducibly occurred. Thus, avicin A appears to exhibit a bacteriostatic effect in this case, in contrast to the bactericidal action of acidocin A.

The truncated analogs of acidocin A showed reduced activity compared to the full-sized peptide. The degree of reduction differed for different combinations of tested fragments and test strains; however, no obvious advantage of any fragment over the others was observed. In particular, both the *N*-terminal (1–31) and *C*-terminal (32–58) fragments of the original sequence demonstrated partial retention of antimicrobial properties against some strains.

Designing hybrid molecules consisting of fused parts of different AMPs makes it possible in some cases to obtain artificial peptides with new activity spectra that are a combination of the activity spectra of the parent molecules. We tested two variants of coupling the *N*-terminal part of avicin A with the *C*-terminal part of acidocin A in the hope of obtaining a molecule with activity against *Listeria* (Appendix A), but did not achieve the desired result. The activity of the obtained hybrids is presented in the Appendix A. Apparently, specific interaction with Man-PTS is not possible with such major changes in the structure of the molecule.

### 2.5. Antifungal Activity

Acidocin A had the greatest antifungal activity of all the studied peptides (Table 4). This peptide effectively inhibited the growth of all the yeast strains tested with MIC of 4 μM in most cases including collection strains and clinical isolates, which are sensitive and resistant to conventional antimycotics. Microscopic analysis showed the presence of yeast cells in wells to which acidocin A was added at a concentration equal to its MIC (Figure 4). At higher peptide concentrations, a decrease in cell number and cell debris was observed, probably due to cell lysis (Figure 4). The peptide had a fungicidal action. For some *Candida* strains, the fungicidal index (FI) was minimal (equal to 1) (Table 4; Appendix A). Among the various strains of *C. albicans*, the resistant strain ATCC 10231 and clinical isolate 9.1 were less sensitive to acidocin A.

Two other peptides, acidocin 8912 and acidocin 8912A, were much less active than acidocin A (Table 4). The MICs of acidocin 8912 and 8912A for most fungal strains were 16 µM and 32 µM, respectively. *C. tropicalis* v13a4/2 was the most sensitive to acidocin 8912A (MIC was only 4 µM). Microscopic analysis also showed that in the case of the most sensitive *Candida* strains, destruction of the fungal cells by acidocin A occurs in the wells with the peptide concentrations exceeding its MICs (which was further confirmed by flow cytometry data, Section 2.7). Acidocin 8912 and acidocin 8912A acted as fungicidal against the most sensitive *Candida* strains (Table 4).

The least active of the tested peptides was avicin A. The MICs of this peptide against most of the tested yeast strains were more than 32 μM (Table 4). Microscopic analysis revealed the presence of numerous yeast cells in the wells with the maximum peptide concentration used (32 µM). At the same time, in the case of *C. krusei* 225/2 and *C. tropicalis* v13a4/2, active destruction of the fungal cells was also observed in the wells with peptide concentration equal to MIC and lower. It is possible that the action of avicin A against the tested *Candida* strains was fungistatic.

A study of the influence of various salts and serum components on the antifungal activity of the four studied peptides was carried out using *C. albicans* ATCC 18804. The antifungal activity of all peptides was significantly reduced in the presence of NaCl or CaCl_2_ at physiological concentrations as well as in the presence of FBS (Appendix A). The MICs exceeded the maximum peptide concentrations used, as follows: 16 μM for acidocin A and 32 μM for acidocin 8912, acidocin 8912A and avicin A. Interestingly, the presence of MgCl_2_ had virtually no effect on the antifungal activity of the acidocin A and acidocin 8912A (Appendix A).

### 2.6. E. coli ML-35p Membranes Disrupting Activity of Acidocins 8912/8912A

The broad spectrum of action of acidocins 8912 and 8912A, their relatively high MICs and toxicity to eukaryotic cells suggest that the mechanism of action of these peptides is the disruption of the barrier function of target cell membranes. In order to test this assumption, the effect of peptides on the membrane permeability of *E. coli* ML-35p in the presence of the chromogenic substrate ONPG was investigated. A number of truncated analogs of acidocin A (AcdA) were also tested. It was shown that the ability to damage the membrane decreases in the row, as follows: melittin > acidocin 8912 > acidocin 8912A > AcdA(32–58) > AcdA(10–48) ≈ AcdA(19–58) ≈ AcdA(1–31) ≈ acidocin A > AcdA(10–39) ≈ AcdA(10–33, C31S) > AcdA(10–33) ≈ avicin A (Figure 5). The high membranotropic activity of acidocin 8912 and acidocin 8912A is apparently due to their ability to form an amphiphilic α-helix at the interface, which is confirmed by CD spectroscopy data (see Section 2.3).

### 2.7. C. albicans ATCC 18804 Membrane Disrupting Activity of Acidocin A

Flow cytometry was used to study the ability of acidocin A to penetrate the cell membrane of *C. albicans* ATCC 18804. The cells were stained with red-fluorescent nucleic acid stain propidium iodide (PI), which can penetrate only dead cells with damaged membranes. Untreated and heat-killed *C. albicans* cells were used as negative and positive controls, respectively. It was shown that acidocin A was able to damage the yeast membrane in all concentrations used after treatment for 2 h. In heat-treated or peptide-treated cells, the increase in PI fluorescence was manifested as a distinct shift of the peak along the *x*-axis (Figure 6B–F, PI vs. count diagrams). In contrast to live cells (Figure 6A) and heat-treated cells (Figure 6B), treatment with acidocin A resulted in a change in the cell morphology (Figure 6C–F, FSC vs. SSC diagrams). Such a pronounced effect can be attributed to the lysis of *C. albicans* cells following the disruption of their membrane integrity by acidocin A. The percentage of PI-stained cells increased from 27% (in the case of treatment with 0.5 × MIC of acidocin A) to 68% (treatment with 4 × MIC) (Figure 6C–F, FSC vs. PI diagrams), indicating that the observed effect is dose-dependent. At the same time, the conventional antimycotic amphotericin B, which has a fungicidal effect against *C. albicans* ATCC 18804 at a concentration of 1 μg/mL (2 × MIC) in our antifungal activity assay, did not show a significant effect on membrane permeability after cell treatment for 2 h. A small number of PI-positive cells were detected only after treatment with 4 × MIC of amphotericin B (Appendix A). Apparently, more time is required for amphotericin B to exhibit its fungicidal effect on *C. albicans* ATCC 18804 cells.

### 2.8. Hemolytic Activity and Cytotoxicity of Acidocins 8912/8912A

Acidocin 8912A showed in experiments a high hemolytic activity against human erythrocytes (HC50 4–8 μM) comparable to that of the membranolytic peptide melittin (Figure 7). The HC50 value determined for acidocin 8912 was significantly higher (≈64 μM), but it still appeared to be a more potent hemolytic than acidocin A in our previous study (HC10 > 128 μM) [19]. The cytotoxic effect of acidocin 8912 against THP-1 and PBMC suspension cell lines was evident at concentrations ranging from 32 μM (Figure 8).

### 2.9. Proteolytic Stability

The resistance of acidocin A to digestion by the major proteases of the gastroenteric tract was monitored by Laemmli SDS-PAGE after 5–120 min of incubation at 37 °C (Appendix A). Densitometric analysis of the gels showed that the peptide was moderately resistant to pepsin (more than half of the initial amount was retained in the reaction mixture after 2 h), but it was almost completely degraded by the duodenal enzymes within the first 5 min of incubation.

### 2.10. Transepithelial Transport of Acidocin A and Avicin A

The Caco-2 cell is an immortalized cell line of human colorectal adenocarcinoma cells. Under specific culture conditions, the Caco-2 cells become differentiated and polarized with intercellular tight junctions, a well-differentiated brush border, and typical small intestinal nutrient transporters, resembling the enterocytes lining the small intestine, making it ideal for intestinal absorption simulations. This is why the Caco-2 permeability assay has become the gold standard method to evaluate both the passive and active transport and the absorption of orally administered drugs and various compounds. Labeling a protein or peptide with FITC is a common approach to evaluate its transport through Caco-2 monolayers [23]. In the current study, we used FITC-labeled acidocin A and avicin A to assess bidirectional “apical-to-basolateral” (A→B, absorptive) and “basolateral-to-apical” (B→A, secretory) transport across Caco-2 polarized monolayers. Mean apparent permeability A→B coefficients (P_app_) were 5.27 × 10^−6^ cm/s for acidocin A and 5.28 × 10^−6^ cm/s for avicin A, while mean P_app_ B→A coefficients were 3.73 × 10^−6^ cm/s for acidocin A and 3.55 × 10^−6^ cm/s for avicin A, which predicts a moderate transepithelial absorption of both bacteriocins in the human gut (Figure 9). The established relationship between P_app_ values and the in vivo absorption of drugs in humans allows us to correlate apparent permeability values between 2 × 10^−6^ and 10^−5^ cm/s with a 20–70% absorption in the gut, which could be expected in humans [24]. An efflux ratio ER = {P_app_(B→A)/P_app_(A→B)} < 1 in the case of both compounds (ER = 0.6–0.7) indicated no involvement of apical efflux transporters in transepithelial transport of both bacteriocins through the Caco-2 monolayer.

## 3. Discussion

Despite the small proportion of identical amino acid residues in the sequences of acidocin A and acidocin 8912, a careful comparison of their primary structure (Figure 1), genetic context and biological activity suggest that these two AMPs share considerable similarity and can be assigned to the same structural family, related to PLB, but with distinctive features. Judging from CD spectroscopy data, acidocin 8912 is definitely closer to acidocin A than to a classical PLB avicin A in terms of polypeptide backbone folding, despite its much smaller size.

The similarity between acidocin 8912 and the *N*-terminal part of acidocin A suggests that evolution proceeded by successive amino acid substitutions rather than by translocation of the entire fragment of the nucleotide sequence. The rate of mutation accumulation was clearly higher in the region between the double glycine site and stop codon, suggesting the action of driving selection. In classical PLBs, the pediocin box plays a key role in the specific interaction with the receptor, the Man-PTS transmembrane carbohydrate transporter [25]. The conservation of its structure could be explained by coevolution with the subunits IIC and IID of the Man-PTS complex [26]. The pediocin box in acidocin A contains one “extra” amino acid and apparently no longer plays a key role in its mechanism of action [19]. In acidocin 8912, it is even further diverged from the canonical sequence.

It is conceivable that the acidocin A-like peptides may have branched off from the PLB evolutionary tree, and together with the loss of the pediocin box integrity, they lost the ability to specifically bind to Man-PTS. Instead, their low-specific membranolytic properties were enhanced and the ability to inhibit the growth of Gram-negative bacteria and fungi appeared. On the other hand, it cannot be ruled out that evolution has proceeded in the opposite direction: from the pool of low-specific membranolytics such as acidocin A, nature could have selected variants with a higher affinity for Man-PTS that were capable of efficiently inhibiting a narrow range of Gram-positive bacteria using these transporter complexes for growth in the environments rich in simple carbohydrates. Due to the lack of data on intermediate forms in the sequence databases, at present, it is not possible to propose a more detailed and valid hypothesis.

In the antibacterial tests, acidocin 8912 showed activity against the same set of strains as acidocin A, except for a few Gram-negative bacteria, but the MIC values for this peptide were several times higher. Approximately the same was the antibacterial activity of acidocin 8912A, a peptide whose amino acid sequence was obtained by combining the sequence of acidocin 8912 with the *C*-terminal acidocin A-like “extension” encoded by the 5-UTR region of its gene, via a glycine residue. Similar to acidocin A and in contrast to PLBs, acidocins 8912 and 8912A lack antilisterial activity. Because of the importance of this fact, it was confirmed by the agar diffusion method. Thus, all three of the above peptides are fundamentally different from most PLBs with regard to the spectrum of their antibacterial activity.

In order to identify the sequence region that makes the main contribution to the antibacterial activity of acidocin A, we obtained a series of truncated analogs of this peptide ranging from 24 to 40 aa in length. Testing of these analogs against bacterial strains sensitive to acidocin A showed that deletion of the *N*- or *C*-terminal parts of the peptide equally resulted in only a partial loss of activity, manifested as a several-fold increase in MICs. Similar results were shown by testing the membranotropic activity of analogs using ONPG. Thus, we were unable to find any key sites within the amino acid sequence of acidocin A.

All the studied peptides in this work have varying degrees of pronounced antifungal activity against pathogenic fungi of the *Candida* genus. The most active peptide is acidocin A, which completely inhibits the growth of various sensitive and resistant strains of *C. albicans*, *C. tropicalis*, *C. krusei* and *C. glabrata* at a concentration of 4 or 8 µM. The peptide has a fungicidal effect, causing lysis of fungal cells at high concentrations.

In contrast to bacteriocins, which bind with high affinity to specific molecular targets in microbial cells [27,28], membranotropic peptides with a broad spectrum of antimicrobial activity are generally able to damage human cell membranes. To our surprise, acidocin A showed no marked hemolytic activity at concentrations as high as 128 µM in the previous study [19]. However, in the present work, acidocins 8912 and 8912A showed a significantly higher ability to lyse human red blood cells (HC50 ≈ 64 µM and 4–8 µM, respectively), which means almost no selectivity of action against bacterial cells for acidocin 8912A. Notably, acidocin 8912 exhibited cytotoxicity against normal (PBMC) and immortalized (THP-1) suspension cell cultures within approximately the same concentration range as acidocin A (32–64 µM).

The wide spectrum of antimicrobial activity of acidocin A and its moderate cytotoxicity make it a promising subject for further research as a possible prototype of a new antibacterial or antifungal agent. Compared to acidocin A, the only obvious advantage of acidocin 8912 so far is its smaller molecule size, which is an important characteristic in terms of biotechnological and pharmacokinetic properties. However, in our opinion, the antimicrobial activities of these peptides should primarily be considered in the aspect of the regulation of the human intestinal microbial community, into which their natural producers can be introduced as a result of the consumption of fermented milk products or probiotics. It should be noted that, in sufficiently high quantities, these peptides have the potential to inhibit both pathogenic and symbiotic microorganisms. Their possible negative effects on the human microbiota have yet to be evaluated.

Activity data obtained in the presence of inorganic salts and proteins and also the results of proteolytic stability assays may raise doubts about the ability of such AMPs to exert their activity under physiological conditions. On the other hand, these data should not be regarded as definite proof to the contrary. Components potentiating the effect of the peptide (such as other AMPs with different mechanisms of action) may be present in biological media. The products of partial proteolysis can form an antimicrobial cocktail that acts more effectively than the original peptide, as has been shown, for example, for human defensins HD-5 and HD-6 from Paneth cells [29]. Finally, much depends on the local concentration of the peptide in the medium, information on which is unfortunately completely lacking at present. Thus, the in vitro test data presented here are only the first step toward understanding the real contribution of bacteriocins in maintaining and normalizing microbial homeostasis.

Lastly, we investigated the bidirectional transport of acidocin A and avicin A across the Caco-2 monolayer used as a simplified model of the human intestinal epithelium. It was shown that both peptides have very similar characteristics and close values of P_app_ for both directions. Based on determined P_app_ coefficients, one can expect that both peptides would have moderate absorption in the human gut (20–70% absorption). This experiment was designed to complement our earlier study on the immunomodulatory activity of these peptides [19]. Of course, this model does not take into account the many factors operating in real-world conditions and, above all, the cleavage of the peptides by the major intestinal proteases. Nevertheless, this pilot experiment once again leads us to speculate that among the peptides of bacterial origin that are permanent components of the human blood peptidome [30], there may be bacteriocins produced by intestinal microbiota (or fragments thereof), which may thus influence not only local but also systemic immune processes.

## 4. Materials and Methods

### 4.1. Search for Acidocin A Homologs

The initial search was conducted using the TBLASTN algorithm (https://blast.ncbi.nlm.nih.gov/Blast.cgi; accessed on 12 February 2024) through NCBI nr/nt database limited to bacteria (taxid: 2) using acidocin A sequence and its fragments as a protein query. The output sequences were analyzed using the BAGEL4 web server (http://bagel4.molgenrug.nl; accessed on 15 February 2024) [31].

### 4.2. Antimicrobial Peptide Preparations

All peptides for this work were obtained using heterologous expression in *E. coli* BL21 (DE3) as fusion proteins with thioredoxin and purified by immobilized metal affinity chromatography (IMAC), CNBr cleavage and reversed-phase high-performance liquid chromatography (RP-HPLC) as described earlier [19]. The amino acid sequences of acidocin A (GenBank BAA07120.1), 8912 (BAA07737.1) and 8912A (Figure 1) were reverse translated using codon optimization tool [32] according to *E. coli* codon usage bias. Coding DNA sequences (CDS) for bacteriocins were synthesized by PCR and inserted into pET-His8-TrxL vector derived from pET-His8-TrxL-Aur [33] downstream of the thioredoxin A (M37L) CDS by in vivo homology recombination. Plasmids encoding the truncated variants of acidocin A and its hybrids with avicin A were obtained by site-directed mutagenesis using full-length plasmid amplification by inverse PCR with mutagenic primers and recircularization by in vivo homology recombination.

The transformed *E. coli* BL21 (DE3) cells were grown up to OD_600_ 0.8–1.0 at 37 °C in lysogeny broth (LB) containing 20 mM glucose, 2 mM MgSO_4_ and 100 mg/mL of ampicillin and then induced with 0.2 mM isopropyl β-D-1-thiogalactopyranoside (IPTG) (Anatrace, Maumee, OH, USA). The cells were cultivated in Erlenmeyer flasks for 4–6 h at 30 °C with shaking at 220 rpm, pelleted by centrifugation, and sonicated in IMAC loading buffer (pH 7.8) containing 100 mM sodium phosphate, 20 mM imidazole and 6 M guanidine hydrochloride (Sigma-Aldrich, St. Louis, MO, USA). The lysates clarified by centrifugation at 20,000× *g* for 30 min at 4 °C were applied onto a column packed with Ni Sepharose 6 Fast Flow (Cytiva Life Sciences, Marlborough, MA, USA). The recombinant proteins were eluted with the buffer of the same composition supplemented with 0.5 M imidazole. The eluate fractions containing fusion proteins were acidified with 11.5 M HCl (about 54 μL per 1 mL of eluate) to pH~1.0 (monitored with indicator paper) and cleaved by 100-fold molar excess of cyanogen bromide (CNBr) (Sigma-Aldrich, St. Louis, MO, USA) over methionine (50 μL of 50% (*m*/*V*) CNBr solution in acetonitrile per 1 mL of eluate) for 20 h at 25 °C in the dark. The reaction products were lyophilized, dissolved in water, titrated to pH ~5.0 and loaded onto a semi-preparative Reprosil-pur C18-AQ column (250 mm × 10 mm, particle size 5 µm, pore size 120 Å) (Dr. Maisch GmbH, Ammerbuch-Entringen, Germany). RP-HPLC was performed at a flow rate of 2 mL/min in a linear gradient of solvent B (80% acetonitrile; 0.1% trifluoroacetic acid) in solvent A (5% acetonitrile; 0.1% trifluoroacetic acid): 0–30% (*v*/*v*) of solvent B for 15 min; 30–60% for 30 min; 60–100% for 8 min. The peaks were monitored at 214 and 280 nm. The collected fractions were analyzed by MALDI-TOF mass-spectrometry using Reflex III instrument and flexAnalysis 3.3 software (Bruker Daltonics GmbH & Co. KG, Bremen, Germany) and by Tris-tricine SDS-PAGE. The fractions containing the target peptides were lyophilized and dissolved in water. The peptide concentrations were estimated using UV absorbance at 280 nm.

The peptide samples for cytotoxicity and intestinal permeability assays were repurified on a separate Reprosil-pur C18-AQ column under the same conditions as above.

Melittin (>98% pure) was synthesized using a standard solid-phase method in M.M. Shemyakin and Yu.A. Ovchinnikov Institute of Bioorganic Chemistry of the Russian Academy of Sciences (Moscow, Russia).

### 4.3. Circular Dichroism Spectroscopy

Circular dichroism spectra of acidocins in water solution and in detergent micelles of 20 mM dodecylphosphocholine (DPC) (Anatrace, Maumee, OH, USA) and 20 mM sodium dodecyl sulfate (SDS, Sigma-Aldrich, St. Louis, MO, USA) were recorded using a J-810 spectropolarimeter (Jasco, Hachioji, Tokyo, Japan) at 25 °C in a 0.1 cm path length quartz cell (Hellma GmbH and Co. KG, Mullheim, Germany) in the 190–250 nm range. The final concentration of the peptides was 200 mM. Four consecutive scans were performed and averaged, followed by subtraction of the blank spectrum of the solvent. The CONTIN/LL algorithm (https://sites.google.com/view/sreerama, accessed on 12 June 2023) with an SMP56 set of reference spectra was used for data analysis.

### 4.4. Antibacterial Activity Assay

The antibacterial activity of recombinant acidocins against a number of Gram-positive and Gram-negative bacteria (Appendix A) was determined by the method of two-fold serial dilutions in a liquid medium. The strains were stored frozen at −70 °C in 10% glycerol supplemented with 5% lactose. After two consecutive passages on LB-agar (tryptic soy broth (TSB) (Merck-Millipore, Burlington, MA, USA) agar for *Listeria*), they were inoculated into 5–10 mL of liquid LB medium (TSB for *Listeria*) and grown for 16–18 h at 37 °C to the stationary phase. The aliquots of overnight cultures were diluted 50–100 times with a fresh batch of liquid medium and grown to OD600~1.0. Two-fold serial dilutions of bacteriocins were prepared in sterile 96-well flat-bottom polystyrene microplates (Eppendorf, Hamburg, Germany) (Cat. No. 0030730.011) in a volume of 50 µL of sterile 0.1% bovine serum albumin (BSA) to prevent non-specific sorption of peptides to the surface of the plate wells [34]. Mid-log phase bacterial cultures were diluted with the 2 × Mueller-Hinton broth (MHB) (Sigma-Aldrich, St. Louis, MO, USA) to a final cell concentration of 10^6^ CFU/mL. To determine the activity of bacteriocins against *Listeria* and *Enterococci*, 3% TSB was used instead of MHB. Aliquots of 50 μL of diluted bacterial suspensions were added to each well of the plate, which was then incubated at 37 °C and 950 rpm for 24 h using a thermostatable plate shaker PST-60HL-4 (Biosan, Riga, Latvia). As a negative control, 50 µL of the sterile growth medium (2 × MHB or 6% TSB) was added to wells containing 50 µL of 0.1% BSA. The values of the minimum inhibitory concentrations (MIC) were determined as the minimum concentration of the peptide, at which there was no visible growth of bacterial culture. Incubation for 2 h with 20 µg/mL chromogenic substrate resazurin (Sigma-Aldrich, St. Louis, MO, USA) was used to detect viable bacterial cells if they were difficult to visually detect. To study the effect of inorganic salts on the antimicrobial activity of bacteriocins, an antibacterial activity assay was carried out in an MHB medium supplemented with 0.9% NaCl.

### 4.5. Antifungal Activity Assay

Antifungal activity of the recombinant peptides was studied against collection strains of *Candida albicans* ATCC 18804 and ATCC 10231, clinical isolates of *C. albicans* v47a3 and 9.1, clinical isolates of *C. krusei* 225/2, *C. glabrata* 252/2 and *C. tropicalis* v13a4/2 isolated from the patients with human immunodeficiency virus (HIV) infection and provided by the G.N. Gabrichevsky Research Institute for Epidemiology and Microbiology (Moscow, Russia). *C. albicans* ATCC 18804 and v47a3, *C. krusei* 225/2, *C. glabrata* 252/2 and *C. tropicalis* v13a4/2 were characterized as sensitive to antimycotics yeast strains, while *C. albicans* ATCC 10231 and 9.1 were resistant to azoles. *C. albicans* ATCC 10231 was also resistant to anidulafungin (Appendix A).

The study was carried out in Sabouraud broth using 96-well microplates. Yeast cells at a concentration of 4 × 10^4^ cells/mL were mixed with equal volumes of two-fold dilutions of peptides in water, starting with a maximum concentration of 32 μM. Control wells without peptides were also used. Microplates were incubated at 30 °C for 24 h after that yeast growth was estimated using an inverted microscope and by measuring the optical density at 630 nm. MIC was defined as minimal peptide concentration, inhibiting fungal growth by 100%. To determine whether the peptides act fungicidal or fungistatic, the contents of the wells of the plate with peptide concentrations equal to MIC and higher were sown on Sabouraud agar and incubated at 37 °C for 24 h. MFC was defined as minimal peptide concentration, ensuring the death of all fungal cells. FI was defined as the fungicide index, equal to the MFC/MIC ratio and indicating the severity of the fungicidal action of the peptide. The conventional antimycotic amphotericin B was tested under the same conditions against *C. albicans* ATCC 18804, but in this case, the wells of the microplate were previously blocked with 0.1% BSA. Either 150 mM NaCl, 1.25 mM MgCl_2_, 1.25 mM CaCl_2_ or 10% fetal bovine serum (FBS) was added to Sabouraud broth to estimate the influence of various salts and serum on the antifungal activity of peptides tested.

### 4.6. Membrane Permeability Assay

The ability of acidocins to disrupt the integrity of the inner membrane of *E. coli* strain ML-35p was assessed using chromogenic substrate *o*-Nitrophenyl-β-D-galactopyranoside (ONPG) [35]. The bacterial cultures were grown to stationary phase in 3% TSB medium at 37 °C overnight. The cells were washed three times with cold 10 mM phosphate-buffered saline (PBS), pH 7.4 (173 mM NaCl, 2.7 mM KCl, 10 mM Na_2_HPO_4_, 1.76 mM KH_2_PO_4_) and diluted to the concentration of 3.3 × 10^7^ CFU/mL with 10 mM PBS (pH 7.4), containing an increased concentration of NaCl (1.2%, or 205 mM), and dissolved ONPG (AppliChem GmbH, Darmstadt, Germany). Two-fold serial dilutions of the peptides were prepared in 50 µL of 0.1% BSA in a 96-well plate, and 150 µL of the cell suspension was added to each well. The final concentration of bacterial cells was 2.5 × 10^7^ CFU/mL, and the final concentrations of substrate were 2.5 mM ONPG in the final volume of 200 μL. The absorbance of the hydrolysis products of ONPG was measured spectrophotometrically at 405 nm, using a microplate reader AF2200 (Eppendorf, Hamburg, Germany). Control experiments were carried out under the same conditions with 0.1% BSA instead of peptides. For each peptide, three independent experiments were performed.

### 4.7. Hemolysis Assay

The hemolytic activity of acidocin 8912 and acidocin 8912A was tested against the fresh suspension of human red blood cells (hRBCs) isolated from the blood of two anonymous healthy donors (in triplicate for each), according to the methodology described in our previous study [19]. The hRBCs were washed three times with PBS (pH 7.4). Then, the suspension with a concentration of 8% (*v*/*v*) was prepared. Aliquots of 50 μL of this suspension were added to 50 μL of peptide solutions in 0.1% BSA, serially diluted in the wells of a round-bottom 96-well microplate, starting with 64 μM, to obtain a total volume of 100 μL (final hRBC concentration 4% (*v*/*v*), maximum peptide concentration 32 μM). The plates were incubated for 1.5 h at 37 °C, 1000 rpm and then centrifuged at 3500× *g* for 10 min at 4 °C. Aliquots of 50 μL of the supernatants were transferred to a flat-bottom 96-well microplate, and the release of hemoglobin was detected using an AF2200 microplate reader at 405 nm. The suspension of hRBCs in PBS without peptides and with the addition of 0.1% Triton X-100 (Sigma-Aldrich, St. Louis, MO, USA) solution were used as negative and positive controls, respectively. The percentage of hemolysis was calculated using the following equation:(1)$$Hemolysis\left(\%\right)=\frac{A_{S}-A_{0}}{A_{100}-A_{0}}\times 100\%,$$
where A_S_ is the absorbance of the sample, A_100_ is the absorbance of the positive control with 0.1% Triton X-100 added, and A_0_ is the absorbance of the negative control. Melittin, which causes almost complete hemolysis at concentrations of about 10 μM and higher, was used as a reference peptide.

### 4.8. Cytotoxicity Assay

The cytotoxic effect of acidocin 8912 was determined against PBMCs (human primary peripheral blood mononuclear cells) and THP-1 (human immortalized peripheral blood monocytes) suspension cell lines using resazurin-based cell cytotoxicity assay [36,37]. This reagent was chosen as an alternative to MTT to determine cell culture viability because of its simpler detection methodology (resorufin, the product of resazurin reduction, unlike formazan, is soluble in water) and greater sensitivity (resazurin is reduced by a wider range of enzymes, which is not limited, as in the case of MTT, to mitochondrial dehydrogenases). PBMCs collected from a healthy donor and THP-1 were purchased from the American Type Culture Collection (ATCC PCS-800-011 and ATCC TIB-202, respectively).

The cells were seeded into 96-well plates at 2 × 10^6^ (PBMCs) and 1 × 10^6^ (THP-1) cells per well in RPMI-1640 medium (Capricorn Scientific, Ebsdorfergrund, Germany), supplemented with 10% fetal bovine serum (FBS) (Capricorn Scientific, Ebsdorfergrund, Germany). The plates were kept in the atmosphere of 5% CO_2_ at 37 °C for 24 h. After that, the serial two-fold dilutions of acidocin 8912 in the culture medium were added to final concentrations from 0.25 to 64 μM. After 24 h of incubation with the peptide, resazurin was added at a final concentration of 70 µM, and the plates were kept for an additional 16 h in a CO_2_ incubator. The fluorescence was measured with an AF2200 microplate reader using a 535/595 filter. Untreated cells of both lines were used as negative controls. The cell viability was calculated using the following equation:(2)$$Cell\;viability\left(\%\right)=\frac{F_{sample}}{F_{control}}\times 100\%$$

The membranolytic peptide melittin from honeybee venom was used as a positive control.

### 4.9. Proteolytic Stability Assay

The sensitivity of acidocin A to proteolysis by digestive enzymes was examined by simulation of its gastrointestinal digestion in vitro. Gastric digestion was performed using 50 ng of pepsin per 1 μg of acidocin A in 0.05 M HCl (pH~1.3). For duodenal digestion, 2.7 ng of trypsin and 10 ng of α-chymotrypsin (Sigma-Aldrich, St. Louis, MO, USA) per 1 μg of the peptide were added in 200 mM ammonium bicarbonate buffer (pH~8.0). The concentration of the peptide was 0.5 mg/mL in both cases. The reaction mixtures were incubated at 37 °C for 2 h and the aliquots were taken at 5 min, 30 min and 2 h after the start of incubation, immediately mixed with the sample buffer containing 4% SDS, and heated for 5 min at 80 °C to ensure protease inactivation. The degree of proteolysis was monitored by Laemmli SDS-PAGE in 18%T 3%C polyacrylamide gels by applying a volume of the mixture containing 1.5 μg of the peptide per lane.

### 4.10. Caco-2 Monolayer Permeability Assay

Caco-2 cells (ATCC HTB-37) were cultured in DMEM/F12 medium (Corning Inc., Corning, NY, USA), and supplemented with heat-inactivated FBS and antibiotic–antimycotic solution (Gibco) (Thermo Fisher Scientific, Waltham, MA, USA). Cells were seeded onto gelatin-coated 24-well inserts (PET 0.4 μm pore size, 6.5 mm diameter, 0.33 cm^2^ growth area) at a density of 2.5 × 10^5^ cells cm^−2^. The monolayers were grown for 21 days; the growth medium was refreshed every second day. The integrity of monolayers and polarization state of epithelial cells were checked by measuring transepithelial electrical resistance (TEER) using Millicell ERS-2 Voltohmmeter (Merck-Millipore, Burlington, MA, USA). Caco-2 monolayers with TEER > 400 Ω cm^2^ were used in the transport assay. Labeling of acidocin A and avicin A with fluorescein isothiocyanate isomer I (FITC) and Caco-2 permeability assay have been performed in accordance with previously reported protocols [38]. Lucifer yellow (Sigma-Aldrich, St. Louis, MO, USA) was used as internal integrity control. FITC-labeled peptides were used in a concentration of 5 μM and transepithelial transport in vitro across Caco-2 barriers was carried out for 90 min in 4 independent inserts for each studied transport variant.

### 4.11. Propidium Iodide Uptake Assay by Flow Cytometry

Flow cytometry analysis was performed on a Novocyte 2060R flow cytometer (ACEA Biosciences Inc., San Diego, CA, USA) equipped with blue 488 nm and red 620 nm lasers. The analysis was performed using *C. albicans* ATCC 18804 at a final concentration of 2 × 10^4^ cells/mL in 0.5 × Sabouraud broth. Yeast cells were incubated in shaking conditions at 30 °C for 2 h in a volume of 4 mL in tubes pre-blocked with 0.1% BSA in the presence of acidocin A at concentrations of 0.5 × MIC, MIC, 2 × MIC and 4 × MIC (2, 4, 8 and 16 μM, respectively) or without the addition of the peptide. Amphotericin B (Sigma-Aldrich, St. Louis, MO, USA) was tested for comparison at concentrations of 0.5 × MIC, MIC and 4 × MIC (0.25, 0.5 and 2 μg/mL, respectively). After that, untreated control and treated cells were centrifuged at 1600× *g* for 10 min, resuspended in PBS and stained for 20 min at room temperature in the dark with propidium iodide (Biotium, Fremont, CA, USA) at a concentration of 4 μg/mL. The cells after thermal treatment at 99 °C for 10 min were used as a control of dead cells. The obtained data were processed using NovoExpress Software v. 1.2.4 (ACEA Biosciences Inc., San Diego, CA, USA).

## Figures and Tables

**Figure 1 ijms-25-10059-f001:**
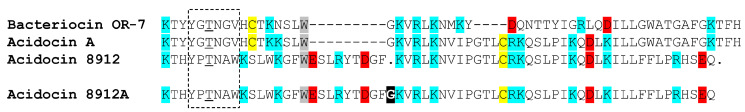
Sequence alignment of bacteriocins of acidocin A subfamily. The region corresponding to the pediocin box is marked with a dashed line; the non-canonical threonine residue is underlined. Positively charged Lys/Arg are highlighted in cyan, negatively charged Asp/Glu residues in red, Cys residues in yellow; conserved tryptophan residues in gray. The sequence of natural acidocin 8912 ends with the Phe residue. Downstream of the first stop codon is a region encoding a sequence homologous to the *C*-terminal part of acidocin A, which is terminated by the second stop codon. Acidocin 8921A obtained in this work comprises the sequences of acidocin 8912 and this *C*-terminal extension, joined by a glycine residue (highlighted with black background).

**Figure 2 ijms-25-10059-f002:**
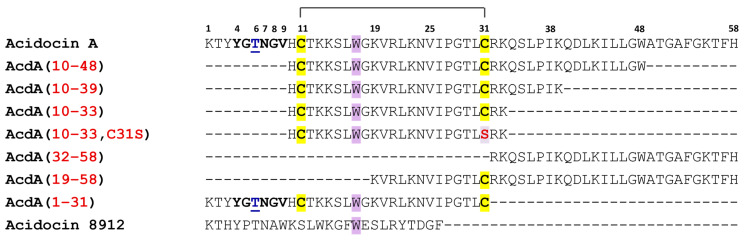
The amino acid sequences of the truncated analogs of acidocin A. Pediocin box is shown in bold; the non-canonical threonine residue is underlined; Cys residues are highlighted in yellow; conserved tryptophan residues in the *N*-terminal part of the molecule in purple; C31S substitution is shown in red. Next to the fragment 1–31, the structure of acidocin 8912 is shown for comparison.

**Figure 3 ijms-25-10059-f003:**
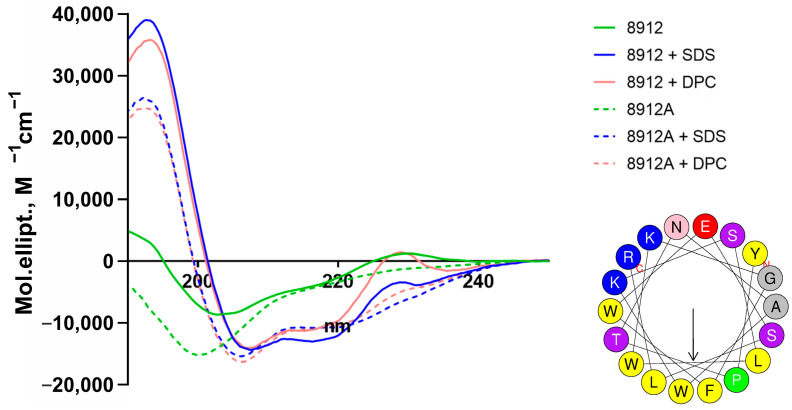
CD spectra for 200 µM acidocin 8912 (solid lines) and acidocin 8912A (dashed lines) in aqueous solution (green) and in the presence of SDS (blue) or DPC (red) micelles. The helical wheel projection (https://heliquest.ipmc.cnrs.fr, accessed on 29 July 2024) of the central part of acidocin 8912 sequence (4–21) clearly demonstrates the spatial segregation of the charged Lys, Arg (blue), Glu (red) and hydrophobic Trp, Leu, Phe (yellow) amino acid residues; the direction of the hydrophobic moment is indicated by the arrow.

**Figure 4 ijms-25-10059-f004:**
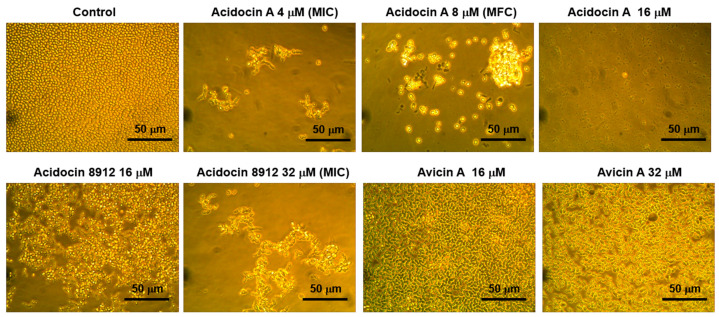
Effects of the peptides at various concentrations on the growth of *C. albicans* ATCC 18804 (×400 magnification).

**Figure 5 ijms-25-10059-f005:**
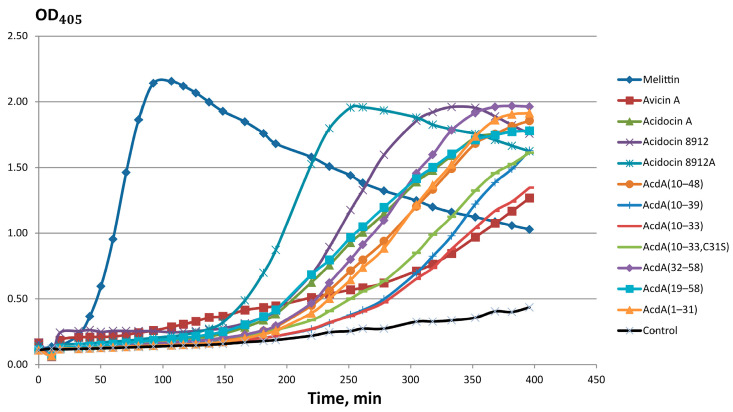
Kinetic curves of *E. coli* ML-35p cytoplasmic membrane permeability in the tests with ONPG. Negative control demonstrates spontaneous hydrolysis of ONPG in the absence of the peptides. The resulting curves are representative of three independent experiments; the shape of the curves for each individual peptide and their relative position to each other was similar in the different experiments.

**Figure 6 ijms-25-10059-f006:**
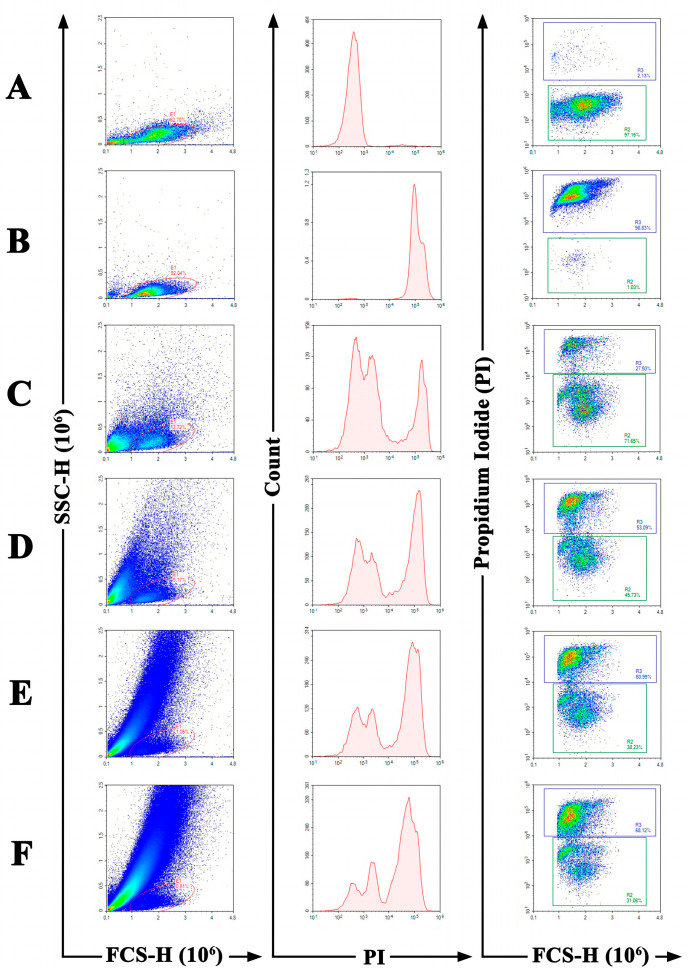
The effect of acidocin A on the cell membrane permeability of *C. albicans* ATCC 18804, measured by PI uptake. Acidocin A at concentrations of 0 (control of live cells, (**A**)), 0.5 × MIC (**C**), MIC (**D**), 2 × MIC (**E**) and 4 × MIC (**F**) were used. Heat-killed *C. albicans* were used as a control of dead cells (**B**).

**Figure 7 ijms-25-10059-f007:**
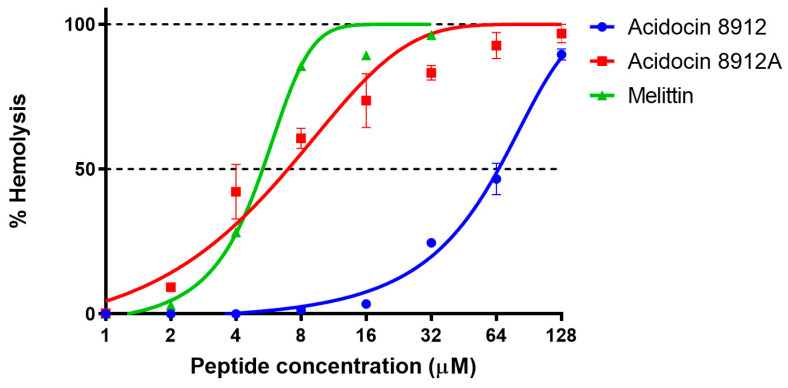
Hemolytic activity of peptides after 1.5 h incubation with human erythrocytes (hemoglobin release assay). Error bars represent standard deviation between technical replications.

**Figure 8 ijms-25-10059-f008:**
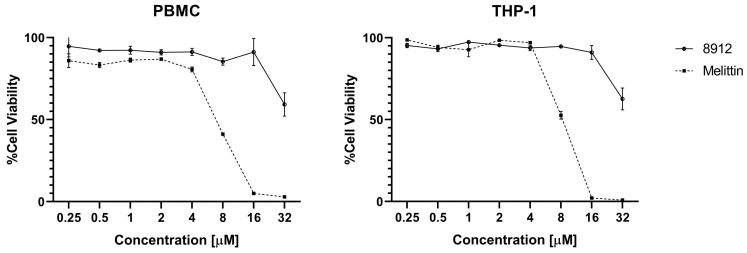
Cytotoxic effect of acidocin 8912 against PBMC and THP-1 suspension cell lines. Melittin was used as a positive control. Error bars represent standard deviation between technical replications.

**Figure 9 ijms-25-10059-f009:**
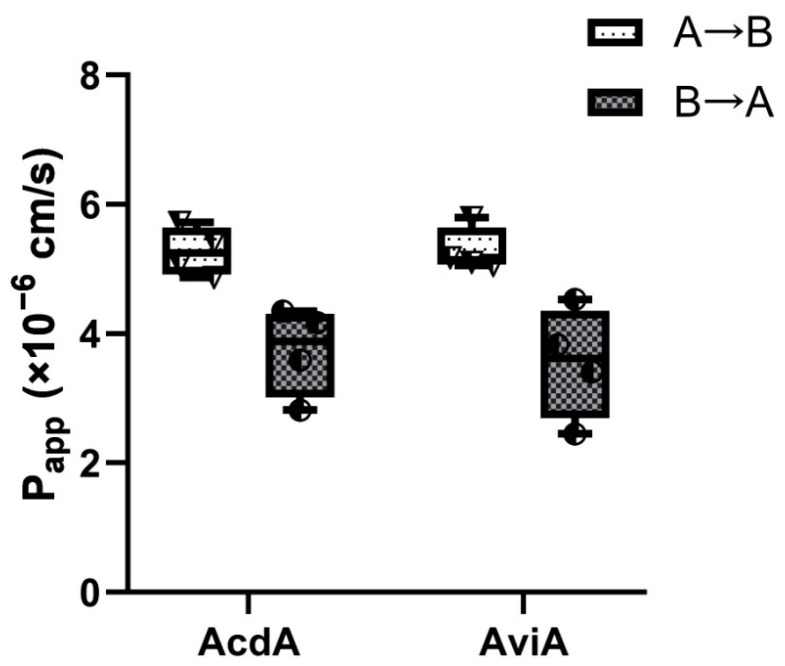
Bidirectional transepithelial transport of bacteriocins acidocin A and avicin A across Caco-2 monolayers.

**Table 1 ijms-25-10059-t001:** Acidocin 8912 and acidocin 8912A secondary structure estimation (%) predicted from far-UV CD spectra compared to previously published data on acidocin A and avicin A [19].

Bacteriocin	Condition	α-Helix, %	β-Sheet, %	β-Turn, %	Random, %	NRMSD
Acidocin 8912	Aqueous solution	4.5	40.5	23.9	31.2	0.05
	DPC micelles	41.4	15.9	22.4	20.3	0.03
	SDS micelles	49.6	4.5	17.0	28.9	0.08
Acidocin 8912A	Aqueous solution	7.9	30.7	23.8	37.5	0.03
	DPC micelles	44.9	7.3	25.2	22.6	0.08
	SDS micelles	44.0	10.1	25.4	20.4	0.04
Acidocin A	Aqueous solution	5.7	33.7	22.6	37.9	0.02
	DPC micelles	44.0	8.1	19.2	28.6	0.01
	SDS micelles	40.2	10.6	20.3	28.8	0.01
Avicin A	Aqueous solution	5.8	32.8	22.1	39.3	0.02
	DPC micelles	19.4	28.6	21.6	27.8	0.02
	SDS micelles	19.7	29.3	23.1	30.4	0.02

**Table 2 ijms-25-10059-t002:** Antibacterial activity of bacteriocins against Gram-positive bacteria.

Bacterial Strains	Minimum Inhibitory Concentration (µM)										
	AcdA	Acd 8912	AcdA (1–31)	AcdA (10–33)	(10–33, C31S)	AcdA (10–39)	AcdA (10–48)	AcdA (19–58)	AcdA (32–58)	AcdA 8912A	AviA
*Listeria monocytogenes* EGD	>32	>32	>32	>32	>32	>32	>32	>32	>32	>32	<0.125
*Lactococcus lactis* ssp. *lactis* bv. *diacetylactis* MK66	0.5	4	4	4	nd	8	2	2	2	nd	>32
*Lactococcus lactis* ssp. *lactis* MK43	1	8	8	16	8	8	8	8	nd	nd	>32
*Lactococcus cremoris* B-1569	4	32	>32	nd	nd	nd	8	nd	>32	nd	>32
*Enterococcus faecium* E19	1	32	nd	16	nd	nd	8	nd	nd	32	>32 *
*Enterococcus faecium* E62	2	16	nd	>32	nd	nd	nd	nd	nd	>32	>32 *
*Enterococcus faecium* E63	2	32	nd	nd	nd	nd	nd	nd	nd	>16	nd *
*Bacillus subtilis* B-886	2	32	8	8	8	8	4	8	>32	32	>32
*Bacillus subtilis* B-2895	4	nd	16	8	8	16	4	8	32	nd	>32
*Bacillus licheniformis* B-511	2	32	8	8	8	8	4	8	>32	8	>32
*Mycobacterium phlei* Ac-1221	2	>32	16	8	4	16	4	16	32	nd	>32
*Mycobacterium* *smegmatis* MC2 155	1	>32	nd	nd	nd	nd	nd	nd	nd	>32	nd
*Micrococcus luteus* Ac-2229	8	16	>32	>32	32	>32	nd	32	>32	nd	>32
*Staphylococcus aureus* 209P	8	nd	>32	>32	>32	>32	nd	nd	nd	nd	>32

* Commentary on the avicin A activity against *Enterococci* is given in the text; AcdA—acidocin A, Acd8912—acidocin 8912, AviA—avicin A, nd—not determined.

**Table 3 ijms-25-10059-t003:** Antibacterial activity of bacteriocins against Gram-negative bacteria.

Bacterial Strains	Minimum Inhibitory Concentration (µM)										
	AcdA	Acd 8912	AcdA (1–31)	AcdA (10–33)	(10–33, C31S)	AcdA (10–39)	AcdA (10–48)	AcdA (19–58)	AcdA (32–58)	AcdA 8912A	AviA
*E. coli* SQ110	2	4	>32	>32	16	>32	4	4	>32	8	>32
*E. coli* ML-35p	2	16	>32	>32	16	>32	>32	8	>32	16	>32
*E. coli* ATCC 25922	4	32	>32	>32	32	>32	32	16	>32	nd	>32
*E. coli* XDR CI 1057	8	32	>32	>32	>32	>32	16	>32	>32	nd	>32
*E. coli* CI 214	8	>32	nd	nd	nd	nd	nd	nd	nd	nd	nd
*E. coli* SBS 1936	8	>32	nd	nd	nd	nd	nd	nd	nd	nd	nd
*Pseudomonas* *aeruginosa* PAO1	16	>32	nd	nd	nd	nd	nd	nd	nd	nd	>32
*Acinetobacter* *baumannii* XDR CI 2675	4	nd	>32	>32	32	>32	nd	8	nd	nd	>32

AcdA—acidocin A, Acd8912—acidocin 8912, AviA-avicin A, nd—not determined.

**Table 4 ijms-25-10059-t004:** Activity of peptides against fungi of the *Candida* genus, which are sensitive and resistant to conventional antimycotics (concentrations in μM).

Fungal Strains	Acidocin A			Acidocin 8912			Acidocin 8912A			Avicin A		
	MIC	MFC	FI	MIC	MFC	FI	MIC	MFC	FI	MIC	MFC	FI
*C. albicans* ATCC 18804 *	4	8	2	32	>32	nd	32	>32	nd	>32	nd	nd
*C. albicans* ATCC 10231 *	8	16	2	16	32	2	32	>32	nd	32	>32	nd
*C. albicans* v47a	4	4	1	16	>32	nd	32	>32	nd	32	>32	nd
*C. albicans* 9.1 *	4	16	4	16	32	2	32	32	1	>32	nd	nd
*C. tropicalis* v13a4/2	4	4	1	16	16	1	4	8	2	>32	nd	nd
*C. krusei* 225/2	8	8	1	32	32	1	>32	nd	nd	>32	nd	nd
*C. glabrata* 252/2	4	4	1	16	16	1	16	16	1	>32	nd	nd

Abbreviations: nd—not determined; *—resistant strain.

## Data Availability

All data presented in this study are available from the corresponding author upon reasonable request.

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
