# Peer review of "Acidocin A and Acidocin 8912 Belong to a Distinct Subfamily of Class II Bacteriocins with a Broad Spectrum of Antimicrobial Activity"

_ijms, 2024, doi:10.3390/ijms251810059_

Round 1

Reviewer 1 Report

Comments and Suggestions for Authors

The manuscript presents several strengths that contribute significantly to the field of microbiology. Firstly, the authors successfully identify a distinct subfamily of class II bacteriocins, expanding our understanding of bacteriocin diversity and evolution. Additionally, they elucidate the structural and functional differences between acidocin A and related peptides, providing valuable insights into their antimicrobial mechanisms.

Moreover, the discovery of acidocin A's activity against Candida species suggests broader therapeutic applications, which is particularly noteworthy given the rising concern over fungal infections. The study's exploration of acidocin A’s proteolytic stability and permeability further enhances its potential as a therapeutic agent. Lastly, the examination of immunomodulatory properties provides a comprehensive assessment of acidocin A’s biological potential.

Overall, I believe this manuscript offers valuable contributions to the field and would be a strong candidate for publication.

Thank you for considering my feedback. I look forward to any further steps in the review process.

In this paper, the authors present a novel antimicrobial peptide such as acidocin A and acidocin 8912 with a broad bacterocidal spectrum activity. However, there are a few points that need to be addressed before considering it for publication.

1. The numbering of the subheadings in section "4. Materials and Methods" needs to be corrected. For example, "2.1. Search for Acidocin A Homologs" should be changed to "4.1. Search for Acidocin A Homologs."

2. Additionally, the title "3. Discussion" on page 11 should be written in bold font.

Author Response

1. Summary

Thank you very much for taking the time to review this manuscript. Please find the detailed responses below and the corresponding corrections highlighted in the re-submitted files.

2. Point-by-point response to Comments and Suggestions for Authors

Comments 1: The numbering of the subheadings in section "4. Materials and Methods" needs to be corrected. For example, "2.1. Search for Acidocin A Homologs" should be changed to "4.1. Search for Acidocin A Homologs."

Response 1: Thank you for pointing this out. The numbering has been corrected.

Comments 2: Additionally, the title "3. Discussion" on page 11 should be written in bold font.

Response 2: Agree. Corrected.

Reviewer 2 Report

Comments and Suggestions for Authors

Acidocin A and Acidocin 8912 Belong to a Distinct Subfamily of Class II Bacteriocins with a Broad Spectrum of Antimicrobial Activity

The authors compare 2 bacteriocins of the same family. The results are interesting and the activity against gram-negative bacteria is not yet widespread. However, the study lacks the comparison with other bacteriocins and citing relevant articles. The introduction should also put the study within a broader context of bacteriocin-research.

Due to this, I recommend major modifications and resubmission of the article. With this, English language and grammar can also be revised since several sentences are not complete.

Below are further comments on the article.

Abstract

Line 24-25: Lower than what? This is not clear in the sentence.

Introduction

Line 39: Repetition (pedicocin box)

Line 50-54: References?

Figure 1: red an purple might be hard to distinguish depending on the individual screen settings.

Line 55-62: May also be considered in the discussion. Figure 1 may in results.

è Introduction should be more extended to different bacteriocins and their differences between each other. Also, what makes the difference should be more highlighted. Activity of a bacteriocin against gram-negative bacteria is somewhat new and unusual and could also be more introduced. Considering the low amount of references in the introduction, there are too many self-citations.

Results

Line 84-88: Have the resulted sequences been curated?

Either use Figure 3 or Table 1, not both. I’d suggest keeping table 1.

Line 161-164: Any explanation why no inhibition was detected in serial dilutions (besides bacteriostatic effect)?

Table 2: Use correct bacterial nomenclature. It is Lactococcus lactis subsp. lactis, Lactococcus lactis subsp. lactis biovar diacetylactis and since recently Lactococcus cremoris.

Table 2: Did you test on spores or vegetative cells of Bacillus sp.?

Figure 4: Am I wrong or do I see some contamination with rods in the pictures for Acidocin  A 4 and 8 uM? Would have been interesting to see fluorescence marked pictures to detect cell permeability.

Line 211-212: General remark to MIC: If the bacteriocin does not inhibit at e. g. 32 uM it means that the MIC is higher. This means, you did not measure at the MIC of the compound. MIC based on e. g. CLSI determines whether a strain is resistant or sensitive. However, as much as I know there are not yet any official values for bacteriocins.

Line 213-218: If talking about lysis in strains you should show it in the images.

Line 224-225: This is the better description related to MICs.

Figure 5: If you repeated the trial 3 times, add error bars in the graph (for the points)

Discussion

Line 312-316: References, comparisons with known evolutionary studies related to bacteriocins?

Line 323-327: why could this happen?

Line 344-352: It is not that clear what the authors want to say

Line 368-373: What about acidocin 8912? I would discuss this shortly as well.

Line 374-385: One should also consider negative effects on human microbiota. Alteration of it may also be negative. Bacteriocins are a tool of bacteria to compete against others occupying the same niche.

Materials and Methods

Make sure to name manufacturing companies for all chemicals and equipments. Also, reference the used PCR primers.

How were the strains kept and prepared?

Ethical agreement for human blood donors? Not required for this?

Line 401-404: When were they accessed?

Line 423-424: How where they clarified?

Line 427-427: Explain this better. Also 100% methionine or a solution?

Line 431-432: How was the gradient? According to the supplementary figure it is nor linear (e. g. from 0-100 % in 100 min).

Line 450-452: Why not show the error bars/area?

Line 529-531: Use equation on a separate line and format it using equation editor of word or similar

Line 556: Equation format as above

Line 565-568: Have the proteases been inactivated prior analysis?

Comments on the Quality of English Language

see above. 

Author Response

1. Summary

Thank you very much for taking the time to review this manuscript. Please find the detailed responses below and the corresponding corrections highlighted in the re-submitted files.

2. Point-by-point response to Comments and Suggestions for Authors

Comments 1: Line 24-25: Lower than what? This is not clear in the sentence.

Response 1: Thank you for pointing this out. Corrected.

Comments 2: Line 39: Repetition (pedicocin box).

Response 2: Agree. Corrected.

Comments 3: Line 50-54: References?

Response 3: Added the reference.

Comments 4: Figure 1: red an purple might be hard to distinguish depending on the individual screen settings.

Response 4: Changed purple to gray.

Comments 5: Line 55-62: May also be considered in the discussion. Figure 1 may in results.

Response 5: We agree that this part could have been placed in the Discussion, but because it was the starting point for the whole study, we decided to keep it in the Introduction. We have added a reference to Figure 1 in the Discussion.

Comments 6: Introduction should be more extended to different bacteriocins and their differences between each other. Also, what makes the difference should be more highlighted. Activity of a bacteriocin against gram-negative bacteria is somewhat new and unusual and could also be more introduced. Considering the low amount of references in the introduction, there are too many self-citations.

Response 6: Added some general information on bacteriocins, their classification and examples of peptides active against Gram-negative bacteria.

Comments 7: Line 84-88: Have the resulted sequences been curated?

Response 7: As far as we understand, these sequences were not curated by NCBI staff. The annotation in some cases was provided by the authors, in other cases it was done by NCBI automatic prokaryotic genome annotation pipeline.

Comments 8: Either use Figure 3 or Table 1, not both. I’d suggest keeping table 1.

Response 8: In our opinion, Figure 3 and Table 1 complement each other. We agree that the table is more informative, but the figure, in addition to CD spectra, contains helical wheel projection of acidocin 8912, so it seems more reasonable to keep both in the article.

Comments 9: Line 161-164: Any explanation why no inhibition was detected in serial dilutions (besides bacteriostatic effect)?

Response 9: We note that inhibition of enterococci actually occurred, but for avicin A it was observed only at the early stage of incubation (wells were overgrown after 24 h). This may be why many authors prefer to demonstrate the activity of classical pediocin-like bacteriocins using time-kill curves rather than MIC measurements. We have some speculations on this, but we feel it is somewhat premature to add them to the article.

Comments 10: Table 2: Use correct bacterial nomenclature. It is Lactococcus lactis subsp. lactis, Lactococcus lactis subsp. lactis biovar diacetylactis and since recently Lactococcus cremoris.

Response 10: Thank you for useful remark. We’ve corrected the nomenclature (we have used the common abbreviations ssp. and bv. to save space).

Comments 11: Table 2: Did you test on spores or vegetative cells of Bacillus sp.?

Response 11: As described in the Materials and Methods section, for each strain, we used a fresh mid-log bacterial culture to set up the tests, which, in the case of bacilli, should contain predominantly vegetative cells.

Comments 12: Figure 4: Am I wrong or do I see some contamination with rods in the pictures for Acidocin  A 4 and 8 uM? Would have been interesting to see fluorescence marked pictures to detect cell permeability.

Response 12: As far as we can tell, it's not contamination, but cell debris. The microscope is inverted, but it focuses only at the bottom, while some Candida cells and their fragments are in suspension, so they may give shadows. To demonstrate cell permeability, we conducted a flow cytometry study using the DNA-binding fluorescent dye (Sections 2.7 and 4.11).

Comments 13: Line 211-212: General remark to MIC: If the bacteriocin does not inhibit at e. g. 32 uM it means that the MIC is higher. This means, you did not measure at the MIC of the compound. MIC based on e. g. CLSI determines whether a strain is resistant or sensitive. However, as much as I know there are not yet any official values for bacteriocins.

Response 13: We’ve changed this statement by linking the results of microscopic analysis with new data obtained by flow cytometry in the experiment with acidocin A.

Comments 14: Line 213-218: If talking about lysis in strains you should show it in the images.

Response 14: The evidence of cell lysis was observed in the pattern obtained by flow cytometry study, which we have added to this work (Sections 2.7 and 4.11).

Comments 15: Line 224-225: This is the better description related to MICs.

Response 15: Thank you for pointing this out.

Comments 16: Figure 5: If you repeated the trial 3 times, add error bars in the graph (for the points)

Response 16: In our view, this would clutter the picture and made it difficult to perceive the results as the bacterial growth rates and the response to the addition of the peptides differed markedly on different days. In this assay, attention should be paid to the reproducibility of the relative positioning of the curves rather than to the dispersion of data for each individual peptide between experiments.

Comments 17: Line 312-316: References, comparisons with known evolutionary studies related to bacteriocins?

Response 17: Certainly such a comparison would be very appropriate here. Unfortunately, we were unable to find publications that were relevant to our case.

Comments 18: Line 323-327: why could this happen?

Response 18: Thank you, this is an interesting question. We can only guess what changes in the habitat conditions of bacteriocins gene carriers might have provoked selection in one or another direction. But now, in the absence of data, we would prefer not to engage in speculations. We would like to limit ourselves to pointing out the apparent relationship between the structures, but the direction of evolutionary processes is still beyond our scope.

Comments 19: Line 344-352: It is not that clear what the authors want to say

Response 19: We have rewritten this paragraph.

Comments 20: Line 368-373: What about acidocin 8912? I would discuss this shortly as well.

Response 20: We have added our thoughts on acidocin 8912 in this paragraph.

Comments 21: Line 374-385: One should also consider negative effects on human microbiota. Alteration of it may also be negative. Bacteriocins are a tool of bacteria to compete against others occupying the same niche.

Response 21: Thank you for pointing this out. We have added this consideration to the text.

Comments 22: Make sure to name manufacturing companies for all chemicals and equipments. Also, reference the used PCR primers.

Response 22: Added information on manufacturers of the key reagents and equipment.

Comments 23: How were the strains kept and prepared?

Response 23: We have added the details to Material and Methods.

Comments 24: Ethical agreement for human blood donors? Not required for this?

Response 24: The ethical agreement is not required since the blood samples were obtained from anonymous donors through an organization that does not provide personal data to the investigator. We have changed the text accordingly.

Comments 25: Line 401-404: When were they accessed?

Response 25: We have added this information.

Comments 26: Line 423-424: How where they clarified?

Response 26: Centrifugation conditions have been added to the Materials and Methods section.

Comments 27: Line 427-427: Explain this better. Also 100% methionine or a solution?

Response 27: Missing details have been added to Materials and Methods.

Comments 28: Line 431-432: How was the gradient? According to the supplementary figure it is nor linear (e. g. from 0-100 % in 100 min)

Response 28: Thank you for the clarifying question. We have added the solvents formulations and the gradient profile to Materials and Methods.

Comments 29: Line 450-452: Why not show the error bars/area?

Response 29: The spectropolarimeter used in this work automatically performs repeated measurements and averages the values according to its software settings. We can set the number of repeats, but we do not get raw data for each of them, so we cannot calculate the standard errors.

Comments 30: Line 529-531: Use equation on a separate line and format it using equation editor of word or similar

Response 30: Corrected using MS Word equation editor.

Comments 31: Line 556: Equation format as above

Response 31: Corrected as above.

Comments 32: Line 565-568: Have the proteases been inactivated prior analysis?

Response 32: Yes, this is a critical point. Proteases were inactivated by temperature treatment in the presence of detergent. The conditions have been added to the text.

Round 2

Reviewer 2 Report

Comments and Suggestions for Authors

The authors covered my rrmarks and questions and revised the manuscript accordingly, 

Therefore, the manuscript can be accepted. There are some minor typos in the text which should be corrected prior publishing.

Figure 5: Whereas I agree that error bars might disrupt readibility of the figure, they'd show the range of the values in a biological system. Some of the curves in the figure are actually the same. But the figure can be left as it is.

Some minor additional comments:

Line 431-433: I would reformulate it in this way: A possible negative effect on the human microbiota needs to be evaluated. Or similar.

Line 524-529: Why did you not use MRS which is usually used for Lactobacillus sp.? Not an issue if they grew in LB but they usually grow better in MRS.

Comments on the Quality of English Language

As mentioned above, some typos should be corrected.

Author Response

1. Summary

Thank you very much for additional remarks on our work. Please find the responses below and the corresponding corrections highlighted in the re-submitted file.

2. Point-by-point response to Comments and Suggestions for Authors

Comments 1: There are some minor typos in the text which should be corrected prior publishing.

Response 1: At this time, we have identified and corrected another couple of typos.

Comments 2: Figure 5: Whereas I agree that error bars might disrupt readibility of the figure, they'd show the range of the values in a biological system. Some of the curves in the figure are actually the same. But the figure can be left as it is.

Response 2: Thank you, our view on this issue is the same as yours. In the future, we will try to optimize visualization of the data obtained in such type of assays.

Comments 3: Line 431-433: I would reformulate it in this way: A possible negative effect on the human microbiota needs to be evaluated. Or similar.

Response 3: Thank you for the advice. We have incorporated your wording to this paragraph.

Comments 4: Line 524-529: Why did you not use MRS which is usually used for Lactobacillus sp.? Not an issue if they grew in LB but they usually grow better in MRS.

Response 4: Of course, MRS is the medium of choice for selective growth of Lactobacillus sp. and some close relatives (Pediococci and Leuconostocs, in our experience). However, we did not test our peptides against those species in this work. Among LAB, we included Lactococci to the test panel. And the strains of Lactococci we have in our collection, as we have seen, grow very poorly on MRS.